# A Perspective on the CD47-SIRPA Axis in High-Risk Neuroblastoma

**Xao X. Tang [1], Hiroyuki Shimada [2] and Naohiko Ikegaki [1,\*]**

[1] Department of Anatomy and Cell Biology, College of Medicine, University of Illinois at Chicago, Chicago, IL 60612, USA; xaotang@uic.edu

[2] Departments of Pathology and Pediatrics, School of Medicine, Stanford University, Stanford, CA 94305, USA; hshimada@stanford.edu

\* Correspondence: ikegaki@uic.edu

**Abstract:** Neuroblastoma is a pediatric cancer with significant clinical heterogeneity. Despite extensive efforts, it is still difficult to cure children with high-risk neuroblastoma. Immunotherapy is a promising approach to treat children with this devastating disease. We have previously reported that macrophages are important effector cells in high-risk neuroblastoma. In this perspective article, we discuss the potential function of the macrophage inhibitory receptor SIRPA in the homeostasis of tumor-associated macrophages in high-risk neuroblastoma. The ligand of SIRPA is CD47, known as a "don't eat me" signal, which is highly expressed on cancer cells compared to normal cells. CD47 is expressed on both tumor and stroma cells, whereas SIRPA expression is restricted to macrophages in high-risk neuroblastoma tissues. Notably, high *SIRPA* expression is associated with better disease outcome. According to the current paradigm, the interaction between CD47 on tumor cells and SIRPA on macrophages leads to the inhibition of tumor phagocytosis. However, data from recent clinical trials have called into question the use of anti-CD47 antibodies for the treatment of adult and pediatric cancers. The restricted expression of SIRPA on macrophages in many tissues argues for targeting SIRPA on macrophages rather than CD47 in CD47/SIRPA blockade therapy. Based on the data available to date, we propose that disruption of the CD47-SIRPA interaction by anti-CD47 antibody would shift the macrophage polarization status from M1 to M2, which is inferred from the 1998 study by Timms et al. In contrast, the anti-SIRPA F(ab')$_2$ lacking Fc binds to SIRPA on the macrophage, mimics the CD47-SIRPA interaction, and thus maintains M1 polarization. Anti-SIRPA F(ab')$_2$ also prevents the binding of CD47 to SIRPA, thereby blocking the "don't eat me" signal. The addition of tumor-opsonizing and macrophage-activating antibodies is expected to enhance active tumor phagocytosis.

**Keywords:** high-risk neuroblastoma; macrophages; CD47; SIRPA; SLAMF7; GD2; immunotherapy

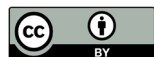

## 1. Introduction

Neuroblastoma is a common pediatric solid tumor of neural crest origin. Neuroblastoma exhibits significant clinical heterogeneity. While some neuroblastomas are low risk and curable, high-risk neuroblastomas progress relentlessly despite the current multimodal high-intensity therapy. The Children's Oncology Group defines high-risk neuroblastomas using several established clinical indicators, including the International Neuroblastoma Risk Group Staging System, age at diagnosis, *MYCN* amplification status, International Neuroblastoma Pathology Classification, the presence or absence of segmental chromosomal aberrations, and ploidy [1]. Based on this classification, about 50% of high-risk cases are *MYCN* amplified, and the rest are non-*MYCN* amplified. Due to the resistance to current therapy, long-term survival of high-risk neuroblastoma patients has remained below 50% for three decades. Moreover, the current therapy for high-risk neuroblastoma causes considerable adverse side effects. Alternative and effective treatments

with low toxicity are needed to treat these patients. Immunotherapy has shown great promise for the treatment of adult tumors [2,3]. However, using immunotherapy as an approach to treat high-risk neuroblastoma has its own challenges. High-risk neuroblastoma cells lack classical human leukocyte antigen (HLA) class I expression [4], and therefore, CD8+ T cell-mediated immune surveillance is considered ineffective for disease control. In addition, the infiltration of NK cells indicated by the expression of NK-specific marker genes is the lowest among immune cells present in high-risk neuroblastoma tissues [5,6]. Consistent with these observations, high-risk neuroblastoma cells do not express the immune checkpoint molecule PD-L1 [5] that controls the activity of PD-1 positive CD8+ T and NK cells [7,8]. Conversely, the majority of PD-L1 positive cells found in high-risk neuroblastoma tissues are macrophages [5]. Subsequently, we have reported macrophages as important immune effector cells in high-risk neuroblastoma [6].

Macrophages are professional phagocytic cells and the major phagocyte population resident in most normal tissues at homeostasis. The activation and phagocytic function of macrophages are tightly regulated to avoid host tissue damage and their self-engagement. Macrophages express a wide spectrum of receptors, which are essential to their immune functions and their vital homeostatic role [9]. Therapeutic strategies for malignant diseases have also exploited the presence of activating and inhibitory receptors on macrophages to promote macrophage-mediated tumor clearance [10].

## 2. Plasticity of Macrophages and Heterogeneity of Tumor-Associated Macrophages

Tumor-associated macrophages (TAMs) arise primarily from bone marrow-derived monocytes, which are recruited to the tumor microenvironment (TME) by the tumor or stroma-derived chemokines [11,12]. As the tumors grow, the TME greatly influences the phenotype of TAMs. The plasticity of macrophages represents their ability to adjust their phenotype in response to various environmental cues, such as cytokines, tissue metabolites, and interactions with other cells [9]. The plasticity of macrophages also gives rise to various subpopulations of macrophages with different phenotypes and functions [9], and the functional effect of the same type of macrophages can be beneficial or detrimental to the host depending on the disease context. Classically activated M1 macrophages exhibit anti-tumor immunity, but they are key mediators involved in immunopathological processes of several autoimmune diseases [9,13]. Alternatively activated M2 macrophages that are important in tissue injury can phagocytose debris, promote wound healing, and antagonize destructive inflammation, but they are pro-tumor in cancers [14]. Although macrophages are relatively abundant in the TME of most solid tumors, they are considered mostly in the immunosuppressive form or with the pro-tumor M2 phenotype [15]. Preserving the anti-tumor M1 TAMs in the TME is thus a key strategy for immunotherapy against solid tumors. In fact, significant efforts have been made in the reprogramming of TAMs from pro-tumor M2 to anti-tumor M1. However, these therapeutic strategies have only shown limited efficacy in human clinical trials against solid tumors [16].

This report is our effort to further explore the role of macrophages as immune effector cells in high-risk neuroblastoma. Below, we will describe the polarization status of macrophages versus their phagocytic capacity in high-risk neuroblastoma at diagnosis and the CD47-SIRPA axis in cancer immunotherapy. In addition, we will present our perspective on (i) the potential function of the macrophage inhibitory receptor signal regulatory protein alpha (SIRPA or SIRP$\alpha$) in the homeostasis of tumor-associated macrophages in high-risk neuroblastoma, (ii) how to optimally enhance tumor phagocytosis by macrophages in high-risk neuroblastoma by targeting SIRPA and activating the macrophages, and (iii) recent clinical trials with anti-CD47 antibodies in human adult and pediatric cancers.

## 3. Macrophage Polarization Status and Phagocytic Capacity in High-Risk Neuroblastoma

The immune microenvironment is unique to individual tumors among high-risk neuroblastomas [5,6], and thus, immunotherapy requires a comprehensive understanding of the composition and phenotypic states of intratumoral immune cells. Of these, macrophages are among the most relevant immune cells in high-risk neuroblastoma because of their predominant presence in the tumor [6]. We were particularly interested in the phenotype and the phagocytic capacity of TAMs in high-risk neuroblastoma. Previously, we reported that most M2-like TAMs could still function as anti-tumor phagocytes in high-risk neuroblastomas at diagnosis [6]. To further explore this, we performed additional analyses on the gene expression profile dataset (*n* = 176) of high-risk neuroblastomas collected at diagnosis [17,18]. As shown in Figure S1, the expression of cytokines, lymphokines, and enzymes for bioresponse modification that influenced the polarization of TAMs were detected in tumor tissues. Furthermore, expression levels of M2 polarization-related genes (*CSF1*, *IL33*, *IL34*, *TGFB1*, *VEGFA, ARG2*) were higher than those of M1 polarization-related genes (*IL1B*, *IL6*, *TNF*, *NOS2*) (Figure S1A). Most hypoxia signature genes reported [19] were also expressed at elevated levels (Figure S1B). These observations suggest that macrophages in high-risk neuroblastoma tissues are in the M2-dominant state. In addition, as shown in Figure S1C, the expression of both *CD68* and *CD163* was significantly correlated with the expression of phagocytosis-related genes [20,21] and macrophage activation marker genes (*CD38* and *CXCL10*) [22,23], suggesting that macrophages in high-risk neuroblastoma tissues at diagnosis retain their phagocytic capacity. Therefore, therapeutic strategies that promote tumor phagocytosis by TAMs beyond diagnosis are essential.

## 4. The CD47-SIRPA Axis in Cancer Immunotherapy

SIRPA is a macrophage inhibitory receptor, which has a restricted expression pattern to myeloid and neuronal cells [24,25]. SIRPA is also known by a variety of names such as BIT, CD172a, MFR, and SHPS-1 [25,26]. SIRPA is expressed in all myeloid cells including monocytes, macrophages, neutrophils, a subset of dendritic cells, and microglia [24,27,28]. The ligand of SIRPA is CD47, which is known as a "don't eat me" signal and is highly expressed on cancer cells compared with normal cells. Based on the current paradigm, the interaction between CD47 on tumor cells and SIRPA on macrophages leads to inhibition of tumor phagocytosis [29–32]. Others have reported the therapeutic efficacy of CD47 blockade with anti-CD47 antibody on various cancers in preclinical models [33–38]. In fact, among the cancer types studied, leukemia is the most susceptible to the anti-CD47 antibody treatment [33,35]. In addition, it has been reported that CD47/SIRPA blockade with either SIRPα variants acting as an analogue of anti-CD47 antibody or anti-SIRPA antibody does not effectively induce phagocytosis on its own and requires additional tumor-opsonizing antibodies to be efficacious [39–41]. Consistent with this notion, it has been shown recently that the CD47/SIRPA blockade therapy with anti-CD47 antibody in xenograft models of high-risk neuroblastoma requires an additional opsonizing antibody against neuroblastoma, such as anti-GD2 antibody, to be efficacious [42]. As will be described below, recent clinical trials with anti-CD47 antibodies have revealed new challenges regarding the CD47-SIRPA axis in immunotherapy.

## 5. The CD47-SIRPA Axis in High-Risk Neuroblastoma

### 5.1. CD47 Expression in High-Risk Neuroblastoma

To determine the cell types expressing CD47 in high-risk neuroblastoma tissues, we performed immunohistochemical analysis on six high-risk neuroblastoma specimens. Our data showed that both stroma cells and tumor cells express CD47. In addition, among the six high-risk neuroblastoma tumors examined, we observed a difference in CD47 staining in high-risk neuroblastoma with *MYCN* amplification vs. those without *MYCN*

amplification. As shown in Figure 1A, among high-risk neuroblastoma with *MYCN* amplification, weak cell membrane staining of CD47 was detected and high CD47 expression was detected on neurites of the tumor cells. In contrast, among non-*MYCN* amplified tumors, high expression of CD47 was detected on both cell membrane and neurites of the tumor cells (Figure 1A). CD47 staining was also detected in the vasculature of high-risk neuroblastoma (Figure S2). For comparison, we examined CD47 expression in favorable histology neuroblastoma. Weak cell membrane staining of CD47 and strong CD47 staining on neurites of the favorable tumor cells were observed (Figure S3).

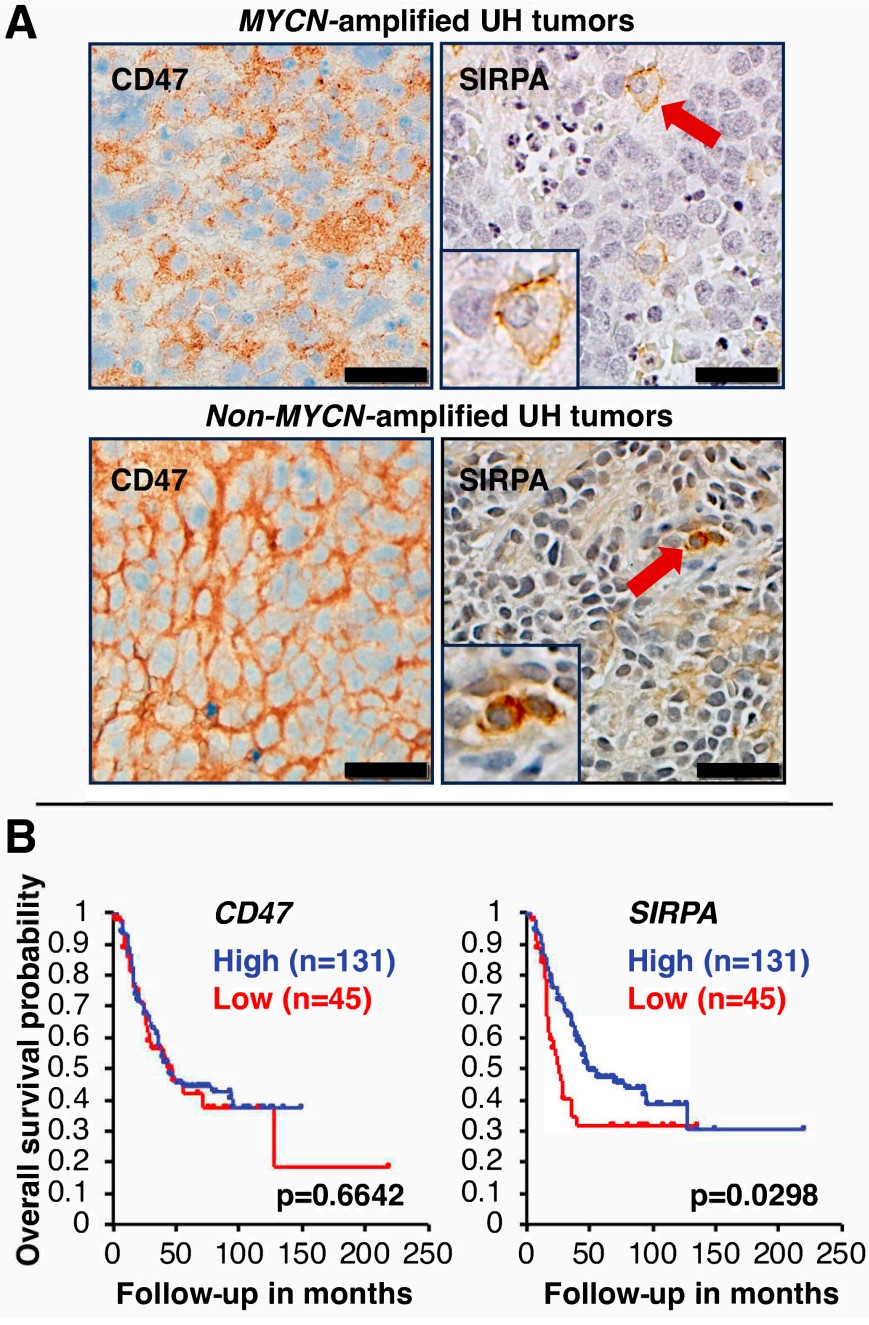

**Figure 1.** (**A**) CD47 and SIRPA expression in high-risk neuroblastoma tissues. Immunohistochemical analysis was performed on high-risk neuroblastoma specimens (three tumors with *MYCN* amplification and the other three tumors without *MYCN* amplification). Representative images are presented. In the high-risk tumor with *MYCN* amplification, weak cell membrane staining of CD47 was detected, and high CD47 expression was detected on neurites of the tumor cells. In contrast, high expression of CD47 was detected on both cell membrane and neurites of the tumor cells without *MYCN* amplification. Some blood vessels were also stained positive for CD47 (see Figure S2). Source

of anti-CD47 antibody: Clone SP279 (Abcam). SIRPA expression was restricted to macrophages in high-risk neuroblastoma tissues. Source of anti-SIRPA antibody: Clone D6I3M (Cell Signaling Technology). Red arrows indicate SIRPA-positive macrophages in tumor tissues. Insets show images of a fully differentiated macrophage (the upper panel) and an early/differentiating macrophage (the lower panel) with 2-fold magnification of the original. Scale bars: 20 μm. UH: unfavorable histology. (B) Clinical implication of *CD47* and *SIRPA* expression in high-risk neuroblastoma. Survival of high-risk neuroblastoma patients with high or low expression of *CD47* or *SIRPA* was analyzed by the R2 Genomics Analysis and Visualization Platform (http://r2.amc.nl, (accessed on 4 November 2023) [43] using the high-risk subset of the SEQC dataset [17,18]. The cohort was split into two groups based on gene expression levels, with the cutoff for the first group set at the first quartile. There was no correlation between *CD47* expression and outcome of high-risk neuroblastoma. In contrast, high *SIRPA* expression was significantly associated with better outcome of high-risk neuroblastoma.

### 5.2. SIRPA Expression in High-Risk Neuroblastoma

We next investigated the expression of SIRPA in high-risk neuroblastoma tissues by immunohistochemical analysis (Figure 1A). In contrast to CD47 expression, among the six high-risk neuroblastoma tumors examined, SIRPA expression was mainly observed in macrophages. Macrophages expressing SIRPA were rather dispersed throughout tumor tissues, but a higher density of smaller SIRPA-positive macrophages at necrotic areas was noticed, suggesting that these macrophages were newly recruited from the circulation in response to tumor-derived chemotactic factors [5,44]. In favorable histology tumor tissues, we also observed SIRPA-positive macrophages (Figure S3).

### 5.3. Prognostic Significance of CD47 and SIRPA Expression in High-Risk Neuroblastoma

To evaluate the prognostic significance of *CD47* and *SIRPA* expression in high-risk neuroblastoma, Kaplan–Meier survival analysis was performed on the high-risk neuroblastoma cohort [17,18]. As described above, CD47 was widely expressed in various cell types (tumor cells and stroma cells) in high-risk neuroblastoma tissues. As expected, there was no correlation between *CD47* expression and disease outcome of high-risk neuroblastoma (Figure 1B). In contrast, the survival analysis showed that 131 cases with high *SIRPA* expression (hazard ratio = 0.61) exhibited significantly better outcome ($p$ = 0.0298) than those ($n$ = 45) with low *SIRPA* expression (hazard ratio = 1.63) (Figure 1B). Based on these data, patients bearing tumors with high *SIRPA* expression had better outcome likely due to more tumor-infiltrating macrophages in the TME, and furthermore, these macrophages were still in the M1-like TAM status and/or M2 TAMs that were able to phagocytose the tumor as suggested by our previous study [6] and Figure S1C.

It should be mentioned that the prognostic significance of SIRPA expression is inconsistent across different cancer types. It has been reported that high SIRPA expression is a favorable factor in colorectal cancer [45], but it is associated with poor prognosis in diffuse large B-cell lymphoma, follicular lymphoma, non-small cell lung cancer, and esophageal carcinoma [46–49]. A plausible explanation for these opposing findings is that the M1/M2 TAM ratio and/or the tumor-phagocytotic capacity of TAMs in colorectal cancer would likely be higher than those of the second group of tumors where high SIRPA expression correlates with poor outcome.

### 5.4. A Missing Piece of the Puzzle

The observations described above led to the following two questions. First, why is there a differential therapeutic efficacy of the CD47/SIRPA blockade in leukemia vs. solid tumors? Second, what is the underlying mechanism for high expression of an inhibitory receptor on macrophages (i.e., *SIRPA*) being associated with a better disease outcome in high-risk neuroblastoma? Notably, the study by Timms et al. [50] published in 1998 sheds light on the answers to these questions. These authors have shown that upon interaction with CD47, SIRPA recruits phosphatases SHP1/2 to its cytoplasmic domain, and SIRPA-SHP1/2 together bind to CSF1R, forming the multi-molecular complex (SIRPA-SHP1/2-CSF1R) to inhibit the CSF1R signaling [50]. To date, CSF1R activation via CSF1 or IL-34 is

known to be required for the M1 to M2 polarization of macrophages [51–55]. The finding by Timms et al. was ahead of its time because the role of CSF1R signaling in M2 polarization was not yet recognized. Because of this, the significance of the original finding by Timms et al. with respect to the macrophage polarization has been left unacknowledged for decades. Consequently, the mechanism underlying the CD47/SIRPA blockade is not fully understood.

### 5.5. A Perspective on the Biological Significance of the CD47-SIRPA Axis in Cancer Immunotherapy and Influences of the CD47/SIRPA Blockade on Macrophage Polarization

Based on the current paradigm, disruption of the CD47-SIRPA interaction by either anti-CD47 or anti-SIRPA antibody would inhibit the "don't eat me" signal and thus promote tumor phagocytosis. However, the current paradigm cannot explain the following. First, the CD47/SIRPA blockade cannot effectively induce phagocytosis on its own when the cancer cells reside in the tumor mass in which TAMs exist. Second, the CD47 blockade is more efficacious in leukemia. Notably, leukemia cells circulate naturally in the blood and will be destroyed by macrophages in the spleen upon administration of the CD47 blockade treatment. To gain further insight into this dilemma and to explain the clinical observation shown in Figure 1B, we incorporate the observation made by Timms et al. [50] and hypothesize that engagement of SIRPA by its partner in some forms (e.g., CD47, anti-SIRPA F(ab')$_2$ lacking FcγR binding) is critical to M1 polarization. Disruption of the CD47-SIRPA interaction by anti-CD47 antibody shifts macrophage polarization from M1 to M2 status, resulting in insufficient tumor phagocytosis (Figure 2C). In contrast, the anti-SIRPA F(ab')$_2$ binds to SIRPA on macrophages, prevents engagement of CD47 with SIRPA, which in turn blocks the "don't eat me" signal. Moreover, engagement of SIRPA by the anti-SIRPA F(ab')$_2$ preserves the capacity of macrophage as M1 TAMs (Figure 2D). Additional macrophage-activating signals via tumor-opsonizing antibodies, such as anti-GD2, would augment active tumor phagocytosis by the M1 TAMs.

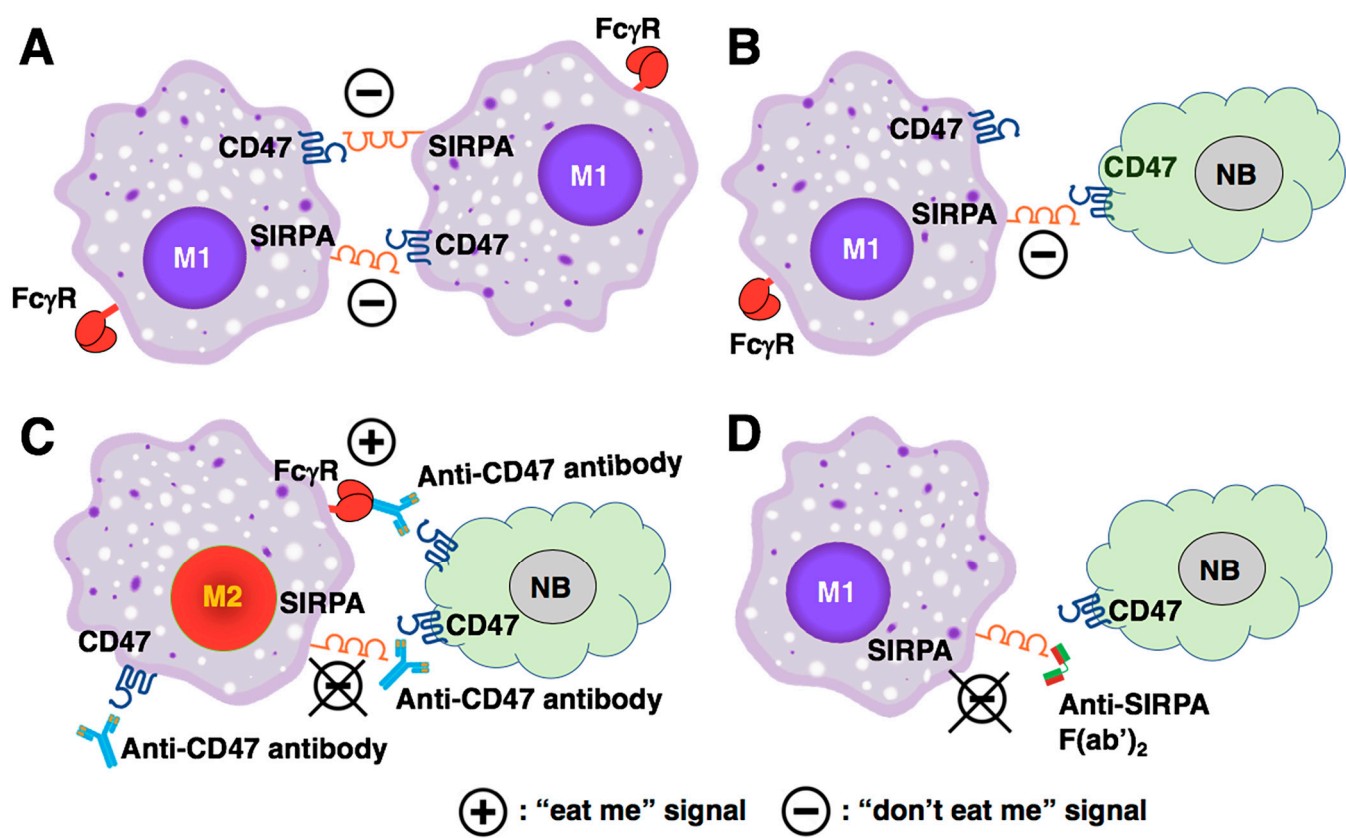

**Figure 2.** Biological significance of the CD47-SIRPA axis in macrophage homeostasis and the influences of CD47/SIRPA blockade on macrophage polarization. (**A**) Under normal physiological conditions, macrophages co-express CD47 and SIRPA. The CD47-SIRPA interaction facilitates macrophages clustering but also prevents macrophages from self-destruction [26]. Moreover, the CD47-SIRPA interaction preserves the phagocytic capacity of M1-like macrophages by inhibiting the CSF1R signaling (see also (**D**)). (**B**) By engaging SIRPA on macrophages, CD47-positive tumor cells send a "don't eat me" signal to the macrophages. (**C**) Anti-CD47 antibody inhibits the "don't eat me" signal. However, disruption of the CD47-SIRPA interaction by the anti-CD47 antibody leads to M2 polarization because the SIRPA engagement is lost. Although anti-CD47 antibody can mediate the tumor killing by ADCP via its binding to Fcγ receptor on the macrophage, the tumor phagocytosis is insufficient due to the M2 TAMs' status. In addition, the anti-CD47 antibody that binds to CD47 on the macrophages can also bind to Fcγ receptors on another macrophage via its Fc portion, which in turn can induce the self-engagement and self-destruction of the macrophages. (**D**) Anti-SIRPA F(ab')₂ binds to SIRPA on macrophage, prevents engagement of CD47 to SIRPA, which in turn blocks the "don't' eat me" signal. Furthermore, engagement of SIRPA by anti-SIRPA F(ab')₂ preserves the capacity of macrophage as M1 TAMs. NB: neuroblastoma.

*5.6. SLAMF7 as a Potential Target to Activate TAMs in High-Risk Neuroblastoma*

It is known that pro-tumor M2 TAMs could persist under the influence of TME-derived TGFβ and glucocorticoid [56–59], for which anti-SIRPA F(ab')₂ cannot block their actions. In fact, *TGFB1* expression was detected in high-risk neuroblastoma tissues (Figure S1A), and most neuroblastomas arise in adrenal glands that produce glucocorticoid. To overcome this, we propose to further activate macrophages by engaging SLAMF7 by F(ab')₂ of the clinically relevant anti-SLAMF7 antibody elotuzumab [60,61]. SLAMF7 is expressed in a variety of immune cells including macrophages [62]. Our previous study [6] and Figure 3 suggest that SLAMF7 is a new therapeutic target for high-risk neuroblastoma. As shown in Figure 3, the immunohistochemistry analysis revealed that SLAMF7 expression was detected on macrophages and a restricted number of high-risk neuroblastoma cells. Moreover, the right image of Figure 3 shows that the SLAMF7-positive high-risk neuroblastoma cells are seen as clusters in the tumor specimen. Within this cluster of the tumor cells, there is a SLAMF7-positive macrophage indicated by the red arrow. The data in Figure 3 are consistent with our previous gene expression analysis which indicates that ~60% of high-risk neuroblastoma likely expressed some levels of *SLAMF7* [6]. Because homotypic interaction is a mode of action of SLAMF7, the expression of SLAMF7 on both macrophages and some high-risk neuroblastoma cells supports the idea that SLAMF7 is a potential therapeutic target for killing the tumor cells.

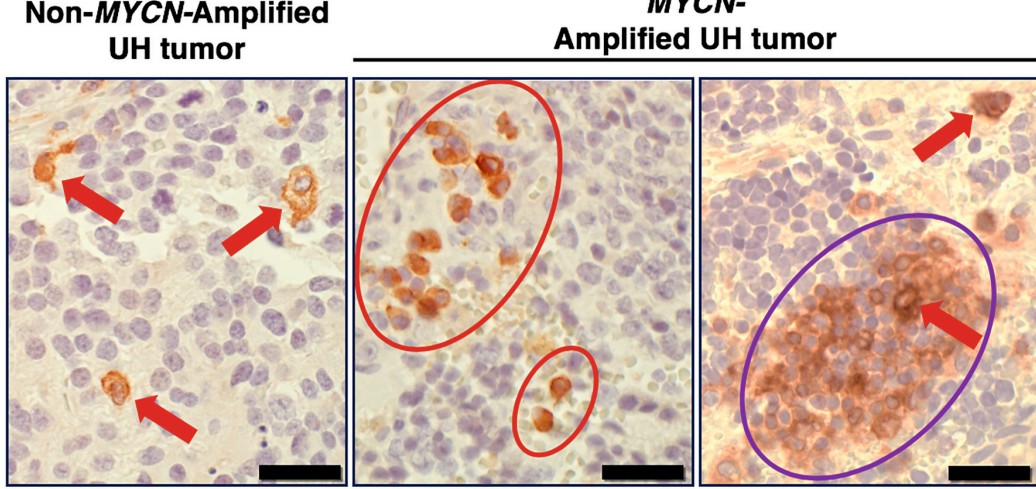

**Figure 3.** SLAMF7 expression in high-risk neuroblastoma tissues. Immunohistochemical analysis was performed on high-risk neuroblastoma specimens (three tumors with *MYCN* amplification and the other three tumors without *MYCN* amplification). Representative images are presented. In the

high-risk tumors examined, SLAMF7 expression was found on macrophages and a restricted number of high-risk neuroblastoma cells. The anti-SLAMF7 antibody, clone EPR21155 (Abcam) was used in the immunohistochemical analysis. Red arrows and red circles indicate macrophages stained by anti-SLAMF7 antibody. The purple circle indicates the SLAMF7-positive high-risk neuroblastoma cells, which are seen as clusters in some tumor specimens. Within this cluster of the tumor cells, there was a SLAMF7-positive macrophage, indicated by the red arrow. UH: unfavorable histology. Scale bars: 20 μm.

*5.7. A Proposed Combination Antibody-Based Therapy for High-Risk Neuroblastoma*

Figure 4 illustrates a proposed antibody-based therapy and its underlying mechanisms to optimally enhance tumor phagocytosis by macrophages in high-risk neuroblastoma. Notably, TAMs in high-risk neuroblastoma tissues highly expressed the macrophage-activating Fcγ receptor genes *FCGR2A* and *FCGR3A* (Figure S1D), and therefore, antibody-mediated approaches are plausible for treating high-risk neuroblastoma patients. As shown in Figure 4, anti-SIRPA F(ab')₂ inhibits the "don't eat me" signal mediated by the CD47-SIRPA axis, blocks M2 polarization, and preserves M1-like TAMs. To further activate macrophages, anti-SLAMF7 F(ab')₂ can be included as an additional therapeutic agent. To directly target the high-risk neuroblastoma cells, the tumor-bound anti-GD2 antibody will be included as a therapeutic of the combination. Anti-GD2 antibody can mediate tumor killing by macrophages via antibody-dependent cellular phagocytosis (ADCP). Anti-GD2 antibodies are currently used in the standard of care for maintenance therapy in patients with high-risk neuroblastoma post autologous stem cell transplant [63]. These include the dinutuximab [64] and humanized 3F8 or naxitamab [65]. In addition, hu14.18K322A is currently being tested in clinical trials [66].

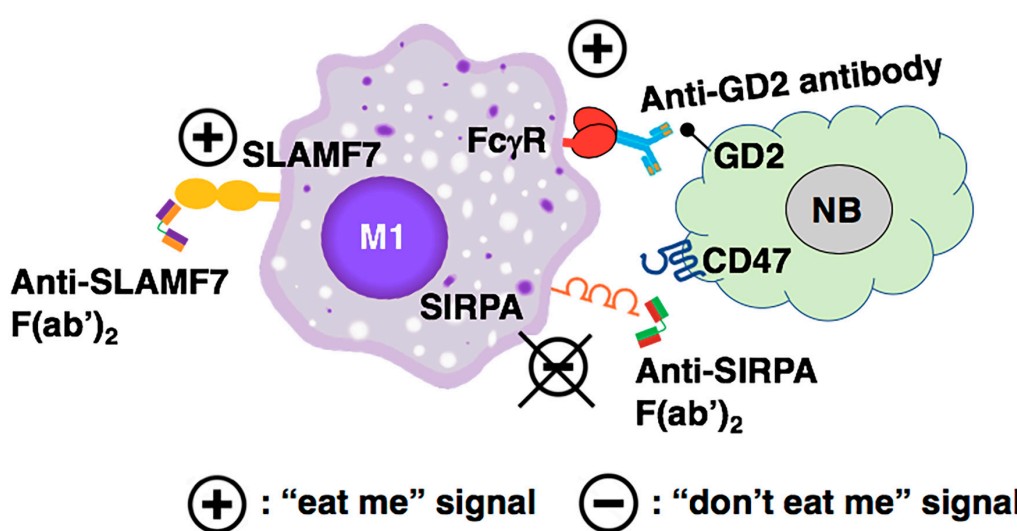

**Figure 4.** The mechanism underlying the proposed antibody-mediated immunotherapy to optimally promote tumor phagocytosis by macrophages against high-risk neuroblastoma cells. Anti-SIRPA F(ab')₂ inhibits the "don't eat me" signal mediated by the CD47-SIRPA axis, blocks M2 polarization, and preserves M1-like TAMs. To further activate macrophages, the anti-SLAMF7 F(ab')₂ can be included as an additional therapeutic agent. To directly target the high-risk neuroblastoma cells, the tumor-bound anti-GD2 antibody is included as a therapeutic of the combination. Anti-GD2 antibody can mediate tumor killing by the macrophages via ADCP.

## 6. Recent Clinical Trials with Anti-CD47 Antibody Magrolimab in Human Adult and Pediatric Cancers

The current development on the first-generation anti-CD47 antibody magrolimab highlights challenges in targeting the CD47-SIRPA axis for cancer immunotherapy. It has been reported that magrolimab and some other anti-CD47 antibodies in clinical trials

cause hemagglutination and platelet aggregation. Furthermore, adverse effects and a lack of survival benefit compared to standard of care have been observed in acute myeloid leukemia (AML) and myelodysplastic syndromes (MDS) trials. Based on these findings, the FDA has placed all magrolimab studies in AML and MDS on full clinical hold (https://www.targetedonc.com/view/fda-halts-clinical-studies-of-magrolimab-in-aml-mds). Gilead Sciences, Inc. the manufacturer of magrolimab (15 February 2024)**,** announced that it has paused enrollment globally in the magrolimab solid tumor studies because FDA requested a partial clinical hold on these trials. For the pediatric oncology, the phase 1 National Cancer Institute-sponsored study (NCT04751383) of magrolimab and dinutuximab (Unituxin) in patients with relapsed/refractory neuroblastoma or relapsed osteosarcoma was also suspended due to unacceptable toxicities (https://clinicaltrials.gov/study/NCT04751383, (accessed on 27 March 2024)). Although cancer cells tend to overexpress CD47, most normal cells, including hematopoietic cells, endothelial and epithelial cells, and fibroblasts (Human Protein Atlas) [62], express CD47 as well, which may have resulted in the observed toxicity of magrolimab therapy.

### 7. Anti-SIRPA Antibodies as Alternatives to Target the CD47-SIRPA Axis

In contrast to CD47, SIRPA expression is restricted to macrophages in many tissues (Human Protein Atlas) [62]. To avoid toxicity of anti-CD47 antibodies due to the ubiquitous expression of CD47, one could target the SIRPA protein on macrophages to inhibit the "don't eat me" signal, block M2 polarization, and preserve M1-like TAMs (Figure 2D). In fact, several anti-SIRPA antibodies have been in development for cancer immunotherapy [67] and tested in human clinical trials (see Table S1). In addition, several clinically relevant anti-SIRPA antibodies are "pan-allelic" and can bind to most SIRPA variants [68–71] existing in human populations [69].

Importantly, when antibodies are used to target specific molecules on the macrophage (i.e., SIRPA in this case), the Fc portion of the antibody could also bind to FcγRs on macrophages. This could in turn cause [1] self-engagement among macrophages, and [2] the competition between anti-SIRPA antibodies and other therapeutic antibodies (e.g., anti-GD2) for FcγR binding, which would dampen the therapeutic efficacy of the tumor-opsonizing antibodies. To circumvent this problem, most clinically relevant anti-SIRPA antibodies have been engineered to include mutations in the IgG1 Fc portion to inhibit or reduce their binding to FcγRs [72]. Some anti-SIRPA antibodies are made with IgG2 and IgG4 isoforms that exhibit a naturally low-affinity binding to FcγRs [73–75]. Alternatively, the use of anti-SIRPA F(ab')$_2$ lacking the Fc portion as a therapeutic agent would be a valid option.

In addition to anti-SIRPA F(ab')$_2$ and the engineered anti-SIRPA antibodies mentioned above, a recombinant CD47 extracellular domain (ED) targeting SIRPA on macrophages could be an alternative therapeutic. However, the natural affinity of CD47 ED against SIRPA is relatively low (a low μM $K_D$ range) [76]. Therefore, a free form of the native CD47 ED (not expressed on the cell surface as multivalent forms) would not be suitable for therapeutic applications. Ho et al. have addressed this problem and identified several CD47 ED variants with increased affinities against SIRPA by several hundred-fold [77]. To this end, a high-affinity CD47 ED-silent Fc fusion protein may also be an alternative therapeutic agent to target the CD47-SIRPA axis and effectively promote macrophage-mediated tumor cell phagocytosis.

### 8. Discussion

Macrophages were discovered as primary immune cells that are always present and ready to engulf and digest microorganisms, providing the first line of defense against infection. To date, macrophages are emerging as an attractive target and focus of cancer immunotherapy due to their relative abundance in the TME of solid tumors, their plasticity, and the numerous receptors expressed on their cell surface [78]. In this article, we

discuss our perspective on the CD47/SIRPA blockade therapy in high-risk neuroblastoma and the dual role of the inhibitory receptor SIRPA on macrophages.

As mentioned above, the study by Timms et al. has shown that upon engagement of CD47 with SIRPA on macrophages, the CSF1R signaling is inhibited [50]. To date, CSF1R activation via CSF1 or IL-34 is known to be required for the M1 to M2 polarization of macrophages [51–55]. Therefore, engagement of SIRPA on macrophages by CD47 on tumor cells not only suppresses phagocytosis function of macrophages but also elicits the second signal, which would block M2 polarization and preserve M1 TAMs. In this perspective article, we propose anti-SIRPA F(ab')$_2$, which mimics the engagement of CD47 against SIRPA on macrophages and preserves anti-tumor M1 TAMs by blocking M2 polarization (Figure 2D). Due to the smaller size of its F(ab')$_2$, anti-SIRPA F(ab')$_2$ can penetrate tumor tissues more efficiently [79]. Small molecules such as anti-SIRPA scFv (single-chain fragment variable) can also easily penetrate the tumor tissue. However, in vivo instability issues of scFv have been reported [80]. Although anti-SIRPA antibodies have been used previously as therapeutics against cancers in preclinical models, including a mouse neuroblastoma model [40,41,81], a comparison between the therapeutic efficacy of anti-SIRPA F(ab')$_2$ and the intact anti-SIRPA antibody has not been explored.

Because SIRPA expression is rather restricted to macrophages in many tissues [62], anti-SIRPA F(ab')$_2$ may provide both safety and efficacy. Anti-SIRPA F(ab')$_2$ may also have the advantage over small molecule inhibitors of targeting pathways involved in M2 polarization. For example, PI3K$\gamma$ inhibitors have been considered for preventing M2 polarization by blocking downstream of the CSF1/CSF1R pathway. However, PI3K$\gamma$ is expressed in a variety of immune cells [62], and therefore, these inhibitors could adversely affect the general immune status of the host. Moreover, PI3K$\gamma$ inhibitors could constrain the M1-polarizing CSF2/CSF2R signaling, which also utilizes PI3K$\gamma$ as one of the downstream signaling effectors [82].

## 9. Conclusions

Based on collective data from others and our group, we suggest that disruption of the CD47-SIRPA interaction by anti-CD47 antibody promotes M2 polarization, and counterintuitively, we propose that anti-SIRPA would lead to preserving M1. In addition, we hypothesize that an effective and less toxic cure for high-risk neuroblastoma should target SIRPA on macrophages and not CD47 on the tumor cells. In essence, the proposed approach will focus on how to make the immune cells (i.e., macrophages) eradicate the tumor cells. We have also proposed various combination treatments for high-risk neuroblastoma. The three-component combination treatment includes anti-SIRPA F(ab')$_2$, F(ab')$_2$ of elotuzumab as the anti-SLAMF7 antibody, and an anti-GD2 antibody, such as dinutuximab and naxitamab (Figure 4). We expect that this combination would effectively promote tumor clearance by macrophages in high-risk neuroblastoma and other GD2-positive solid tumors. Nonetheless, it is equally important to investigate the therapeutic efficacy of the two-component combination treatments: (anti-SIRPA F(ab')$_2$ and F(ab')$_2$ of elotuzumab) and (anti-SIRPA F(ab')$_2$ and anti-GD2 antibody). This is because a balance between maximum efficacy and minimum side effects is essential to cancer immunotherapy. The availability of clinically relevant humanized anti-SIRPA antibodies (see Table S1) has made the proposed therapeutic approach feasible since all three antibodies are already in use in clinics to treat cancer patients. Lastly, because of the heterogeneity of high-risk neuroblastoma, we anticipate that antibody-mediated approaches would further require specific combinations with engineered adoptive cell therapies to treat all high-risk neuroblastoma patients.

**Supplementary Materials:** The following supporting information can be downloaded at https://www.mdpi.com/article/10.3390/curroncol31060243/s1: Figure S1. Macrophage polarization and phagocytic capacity in high-risk neuroblastoma tissues; Figure S2. CD47 expression on vascular cells of high-risk neuroblastoma tissues; Figure S3. CD47 and SIRPA expression in favorable

neuroblastoma; Table S1. Anti-SIRPα antibodies tested in human clinical trials for malignant diseases. References [83–91] are cited in the Supplementary Materials.

**Author Contributions:** X.X.T. and N.I. designed the study, generated and analyzed the data, and conceptualized, wrote, and edited the manuscript. H.S. generated and analyzed the IHC data and edited the manuscript. All authors have read and agreed to the published version of the manuscript.

**Funding:** There is no external funding for this study.

**Institutional Review Board Statement:** All procedures performed in studies involving human participants were in accordance with the ethical standards of the institutional and/or national research committee and with the 1964 Helsinki declaration and its later amendments or comparable ethical standards (Stanford University IRB#: 50930).

**Informed Consent Statement:** Informed consent was obtained from all subjects involved in this study under Stanford University IRB#: 45458.

**Data Availability Statement:** Publicly available datasets were analyzed in this study. This dataset is found at NCBI GEO, GSE62564.

**Conflicts of Interest:** There are no conflicts of interest.

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
