# Peer review of "A Perspective on the CD47-SIRPA Axis in High-Risk Neuroblastoma"

_curroncol, doi:10.3390/curroncol31060243_

Round 1

Reviewer 1 Report

Comments and Suggestions for Authors

Overall, this is a nicely done and presented perspective paper. Here is some additional points for consideration:

- I would suggest revisiting throughout to include more references. Some claims/statements referring to scientific evidence that is according to the authors publicly available are not substantiated with references.

-  In addition to the ethical approval statement, the authors should provide more information on the patients analysed in this study

- For the IHC images, the authors say that they are representative images but of which n number?

- Instead of describing in the figure legend how experiments were conducted, I suggest adding a short materials and method section and it should be very clear what data are in house and what is taken from publicly available datasets. Some are clearly indicated as external datasets, but it is confusing throughout 

- In the supplementary materials, the authors show the Log2 expression of several genes - was this done using tissues from patients available to the authors or is this another dataset? If these are tissues that the authors had available, they should also list all the primers that were used for the PCR and a description of the PCR method (extraction, quantification, etc) should be included in the materials and method. In addition, again, what's the n?

Author Response

Responses to Reviewer 1

Comment 1: I would suggest revisiting throughout to include more references. Some claims/statements referring to scientific evidence that is according to the authors publicly available are not substantiated with references.

 Response: We have included additional references to respond to the reviewer’s concern, although without specific suggestions from the Reviewer, it is difficult for the authors to identify all statements and descriptions in the manuscript that require further substantiation with additional references. However, we hope the revision would satisfy the reviewer. The new references cited include: 4,10,11,12,13,14,15, and 78. The corresponding lines in the revised manuscript are 49, 64, 68, 76, 79, 81, and 377. 

Comment 2: In addition to the ethical approval statement, the authors should provide more information on the patients analyzed in this study.

Response: The informed consent statement is added. The characteristics of specimens analyzed is clearly stated as Unfavorable Histology (MYCN amplified and non-MYCN amplified) and Favorable Histology tumors in the manuscript.  Please note that the International Neuroblastoma Pathology Classification, which was used to classify the neuroblastoma specimens into Unfavorable Histology (UH) and Favorable Histology (FH) categories, includes multiple factors, i.e., age at diagnosis, tumor histology and Mitosis/Karyorrhectic index to assess the biological/clinical status of neuroblastomas at diagnosis.  Additional information on MYCN amplification status to the INPC classification further provide essential information on the nature of neuroblastoma specimens.

 Comment 3: For the IHC images, the authors say that they are representative images but of which n number?

 Response: For each category (i.e., UH with MYCN amplification, UH without MYCN amplification, and FH), we have done three cases each. The number of cases examined is the same for CD47, SIRPA, and SLAMF7. To clarify this, we have revised the figure legends of Figure 3 and Figure S3. Regarding CD47 and SIRPA expression in HR NB, the number of cases examined was described in the original text (Lines 145-146) (Line 160), and (Lines 182-183) and in the revised manuscript: (Lines 145-146) (Line 160), (Lines 182-183).

 Comment 4: Instead of describing in the figure legend how experiments were conducted, I suggest adding a short materials and method section and it should be very clear what data are in house and what is taken from publicly available datasets. Some are clearly indicated as external datasets, but it is confusing throughout.

 Response: The gene expression-based analysis, including the survival analysis, was solely done using the gene expression profile dataset via the R2 platform as described in the manuscript. This information is clearly stated in the figure legends.  Per the Journal Instruction, the format of Perspective articles adheres to that of Review articles, and Materials and Methods section is not included. 

 Comment 5: In the supplementary materials, the authors show the Log2 expression of several genes - was this done using tissues from patients available to the authors or is this another dataset? If these are tissues that the authors had available, they should also list all the primers that were used for the PCR and a description of the PCR method (extraction, quantification, etc.) should be included in the materials and method. In addition, again, what's the n?

Response: As stated in the manuscript, we used the publicly available neuroblastoma gene expression dataset. This was described in the manuscript as Data availability statement: Publicly available datasets were analyzed in this study.  This dataset is found at NCBI GEO, GSE62564. To clarify this, we have modified the figure legend of Figure S1 to include the dataset name and GEO ID #. The original dataset was created by gene expression profile analysis of 498 neuroblastoma specimens in total using the RNASeq technique and the expression level of each gene was expressed as RPM (reads per million).  Log2 of RPM was used in the figures. Therefore, there is no PCR nor primer used.  In this dataset, there are 176 cases of HR cases included, which is also stated in the manuscript.

Reviewer 2 Report

Comments and Suggestions for Authors

This paper focuses on novel immunotherapy for high-risk neuroblastoma, particularly emphasizing macrophage polarization and antibody drugs to regulate it. The description is well-organized and provides valuable information to the readers of this journal. I have a few minor comments as follows:

  1. Line 255: Figure 1B appears to be a mistake and should be Figure 2B. Please verify this.
  2. Can you cite in vitro or in vivo experiments that suggest the efficacy of anti-SIRPA antibody and anti-SLAMF7 antibody for high-risk neuroblastoma?

Author Response

Responses to Reviewer 2

Comment 1: Line 255: Figure 1B appears to be a mistake and should be Figure 2B. Please verify this.

 Response: Figure 1B is correctly cited.

 Comment 2: Can you cite in vitro or in vivo experiments that suggest the efficacy of anti-SIRPA antibody and anti-SLAMF7 antibody for high-risk neuroblastoma?

Response: We would like to mention that the study by Bahri et al. reported that the anti-SIRPA antibody enables antibody-dependent phagocytosis of mouse neuroblastoma cells, which is clearly mentioned in the revised manuscript (line 393). This was cited as reference 73 of the first version and reference 81 of the revised manuscript. To our knowledge, there has been no report on the efficacy of anti-SLAMF7 antibody against human neuroblastoma. Such studies are being planned to be performed.